# Locally Injected Autologous Platelet-Rich Plasma Improves Cutaneous Wound Healing in Cats

**DOI:** 10.3390/ani12151993

**Published:** 2022-08-06

**Authors:** Vasileia Angelou, Dimitra Psalla, Chrysostomos I. Dovas, George M. Kazakos, Christina Marouda, Kyriakos Chatzimisios, Zacharenia Kyrana, Evangelia Moutou, Maria Karayannopoulou, Lysimachos G. Papazoglou

**Affiliations:** 1Unit of Surgery and Obstetrics, Companion Animal Clinic, School of Veterinary Medicine, Faculty of Health Sciences, Aristotle University of Thessaloniki, 54627 Thessaloniki, Greece; 2Laboratory of Pathology, School of Veterinary Medicine, Faculty of Health Sciences, Aristotle University of Thessaloniki, 54124 Thessaloniki, Greece; 3Diagnostic Laboratory, School of Veterinary Medicine, Faculty of Health Sciences, Aristotle University of Thessaloniki, 11 S. Voutyra Str., 54627 Thessaloniki, Greece; 4Unit of Anaesthesiology and Intensive Care, Companion Animal Clinic, School of Veterinary Medicine, Faculty of Health Sciences, Aristotle University of Thessaloniki, 54627 Thessaloniki, Greece; 5Laboratory of Agronomy, School of Agriculture, Faculty of Agriculture, Forestry and Natural Environment, Aristotle University of Thessaloniki, 54124 Thessaloniki, Greece

**Keywords:** cat, platelet-rich plasma, wound, metalloproteinases, laser Doppler flowmetry

## Abstract

**Simple Summary:**

Wound healing in cats is a complex process that can be accomplished by first or second intention. Autologous platelet-rich plasma (PRP) is a product from patients’ own blood that has been found to promote wound healing in both humans and animals. The aim of this study was to investigate the efficacy of PRP infiltration in open wounds in cats. It was found that experimentally created wounds infiltrated with PRP showed a more rapid total wound healing and a better perfusion, supporting the use of PRP in open wounds in laboratory cats. In conclusion, PRP is considered a cost-effective and simple method to accelerate open-wound healing in cats.

**Abstract:**

Cutaneous defects in cats are commonly encountered in clinical practice, and healing can be accomplished by first or second intention. Platelet-rich plasma (PRP) is characterized by a plasma concentration containing a large number of platelets in a small volume of plasma. The objective of the present study was to record the efficacy of PRP infiltration in open wounds in laboratory cats. Six wounds were created in the dorsal midline of eight laboratory cats, with the wounds of one side designated as the PRP group and the wounds of the other side as the control group. Wound healing was evaluated by daily clinical examination, planimetry, laser Doppler flowmetry, and histologic examination on days 0, 7, 14, and 25, and by measurement of metalloproteinases (MMPs)-2 and -9 and tissue inhibitor metalloproteinase (TIMP)-1 on days 0, 14, and 25. Based on the results of the present study, the mean time for full coverage with granulation tissue was shorter in the PRP group, the mean contraction and total wound healing percentage were increased compared to the control group, and finally, the perfusion measured with laser Doppler flowmetry was higher in the PRP group during all examination days. In conclusion, this is the first study focusing on the topical application of PRP in the treatment of open wounds in laboratory cats, and our results are encouraging—showing a more rapid healing in the PRP group.

## 1. Introduction

In cats, skin wounds are usually a result of traumatic injuries, oncologic procedures, or skin infections, and healing may be accomplished by first or second intention. Large cutaneous defects undergoing second-intention healing in cats may result in chronic wounds [1]. Pseudo-healing is also a phenomenon that has been reported in cats; following suture removal from a sutured wound, dehiscence may occur—especially when subcutaneous tissue has been removed [2,3]. It has been found that intact subcutaneous tissue promotes granulation tissue formation in feline cutaneous wound healing [3]. There are published reports that have revealed significant differences, both macroscopically and microscopically, in wound healing between dogs and cats [3,4,5].

Matrix metalloproteinases (MMPs) belong to a family of calcium-dependent and zinc-containing endopeptidases and play a fundamental role in the degradation of the extracellular matrix (ECM) [6,7,8]. Metalloproteinases are classified into groups based on their structure, including collagenases (MMP-1, MMP-8, MMP-13), gelatinases (MMP-2, MMP-9), stromelysines (MMP-3, MMP-10, MMP-11), matrilysines (MMP-7, MMP-26), transmembrane type MMPs (MMP-14, MMP-15, MMP-16, MMP-23), and other MMPs (MMP-12, MMP-19, MMP-20, MMP-21, MMP-23, MMP-27, MMP-28) [9]. Tissue inhibitors of metalloproteinases (TIMPs) are proteins that are secreted and regulate MMP production [7]. The ratio of MMP/TIMP concentration is also considered to be a factor that reflects the progress of wound healing [10]. It has been found in humans that different MMPs are secreted in the different phases of wound healing, with only MMP-2 and MMP-9 expressed constitutively [7]. In small animals, the role of MMPs has been evaluated mainly in dogs and cats with cancer as they are associated with metastasis [11,12,13,14]. There is only one study in which MMP-9 was identified in wound fluid in dogs with full-thickness defects after infiltration with PRP [15]. To the authors’ knowledge, this is the first report evaluating the activity of MMP-2, MMP-9, and their inhibitor TIMP-1 during the wound-healing process in cats after infiltration with PRP.

Platelets play an important role during the wound-healing process—mainly in the first phase of hemostasis-inflammation, although they contribute to all phases of healing [16]. They contain granules that release growth factors that have an important role in the healing process [17]. The most important growth factors that are released from activated platelets include platelet-derived growth factor (PDGF), transforming growth factor-beta (TGF-β), vascular endothelial growth factor (VEGF), insulin-like growth factor (IGF I/II), and epidermal growth factor (EGF) [17]. All these factors have a fundamental role in fibroplasia, angiogenesis, and extracellular matrix formation [18].

Platelet-rich plasma (PRP) is a plasma concentration usually derived from the patient’s own blood (autologous) that comprises predominantly platelets and growth factors, promotes wound healing in soft and hard tissues, and decreases postoperative infection, pain, and blood loss [19]. Platelet-rich plasma contains a large concentration of platelets above the baseline in a small volume of plasma or 2 –5-fold compared with that of whole blood [20,21,22]. This concentration is related to the secretion of growth factors that promote the healing process [23,24,25]. Platelet-rich plasma also includes red blood cells (RBCs) and white blood cells (WBCs). Their increased concentration is not considered to be beneficial as they may stimulate inflammatory factors that can lead to delayed healing [26]. The ideal concentration of these factors in PRP is not well established, and it depends on the tissue on which it is applied [27]. There are also different commercially available methods to produce PRP which differ in the process, the time and speed of centrifugation, and the number of platelets, RBCs, WBCs, and growth factors that are released. The biological effect of PRP is achieved when platelets are activated, resulting in the degranulation of their α-granules and the release of growth factors [17]. Activation can be achieved either with the use of bovine thrombin to activate the clotting process or with the use of calcium chloride to activate the formation of autogenous thrombin [28]. Activation can also be achieved by the production of autologous thrombin during the contact of PRP with the endothelium that has been ruptured and collagen [28,29]. However, the early activation of platelets is not desirable because it can lead to loss of growth factors before the application of PRP [30].

Platelet-rich plasma has been widely and safely used in both animal and human patients. It has been used in dermatology, orthopedics, neurosurgery, ophthalmology, dentistry, and in wound management and plastic surgery [31,32,33,34,35,36,37]. In veterinary medicine, there are experimental and clinical studies reporting the effective use of PRP in rabbits, rats, horses, goats, pigs, sheep, dogs, and cats [27,38,39,40,41,42,43,44,45,46,47,48].

In cats, there are only two case reports for the use of PRP in skin wounds [49,50]. The aim of the present study was to investigate the effect of locally injected autologous platelet-rich plasma on the second-intention wound-healing of cutaneous defects in cats. Our hypothesis was that the application of PRP accelerates cutaneous wound healing in cats.

## 2. Materials and Methods

### 2.1. Animals

Eight neutered, domestic shorthaired (DSH), purpose-bred laboratory cats, five males and three females, aged 1–4 years and weighing 3–4 kg entered the study. A physical examination, complete blood count, and biochemical analysis were performed on all animals. All cats were also screened for the detection of antibodies to feline immunodeficiency virus (FIV) and for the detection of antigens to feline leukemia (FeLV) virus in feline serum. They were housed in individual large cages and were provided with commercial dry maintenance diets twice daily, and water was offered ad libitum. They were accustomed to their new environment for at least one week before the commencement of the study. During this period, a padded bandage was placed on each cat extending from the cranial thoracic to the caudal lumbar region. This experimental study was approved by the State Veterinary Authorities (certificate no. 632,977-3371/16.01.2019) and by the committee for bioethics and deontology of the School of Veterinary Medicine (certificate no. 598/5.8.2019 EL54BIO18). The use of laboratory animals in the present study was in accordance with the Animal Welfare Act and the NRC Guide for Care and Use of Laboratory Animals. Power analysis was performed to allow for an ethically acceptable study using the minimal number of animals necessary to achieve the scientific aims.

### 2.2. Anesthesia

On day 0, the cats were premedicated with acepromazine (0.05 mg/kg of body weight) (Acepromazine; Alfasan, Woerden, The Netherlands) and buprenorphine (0.02 mg/kg) (Bupredine multidose; Dechra Academy, Oudewater, The Netherlands) which were administered intramuscularly (IM). Cephazolin (Vifazolin; Vianex, Athens, Greece) was administered intravenously (IV) at a dose of 20 mg/kg. Anesthesia was induced with propofol (4 mg/kg) (Propofol MCT/LCT; Fresenius Kabi Hellas, Athens, Greece) IV and maintained with isoflurane (2%) (Isoflurane-Vet; Merial, Milano, Italy) in 100% oxygen after endotracheal intubation. Lactated Ringer’s (LR’s; Vioser, Trikala, Greece) solution was administered at 5 mL/kg/h.

### 2.3. Preparation of Autologous PRP

Before the induction of anesthesia, the neck of each cat was clipped for blood collection, and 12 mL of whole blood was collected in a 20-mL syringe via jugular venipuncture using a 22-gauge butterfly. An aliquot of 1 ml of this blood was inserted into an EDTA tube and used for complete baseline hematology in an automated hematologic analyzer (Siemens Bayer ADVIA 120 Hematology System). A blood smear was also performed for all samples to confirm the number and morphology of platelets. The remaining 11 mL was used for the preparation of autologous PRP in a single-spin system using a human kit (Tropocells, Estar medical, Israel). The blood was inserted into a vacuum tube containing separation gel and anticoagulant (MACD7), and it was gently suspended four times before the centrifugation. The anticoagulated blood was then centrifuged for 10 min at 1500 rpm. During this process, the gel separated platelets from platelet-poor plasma (PPP), red blood cells, and granulocytes. In this way, the platelets remained on top of the gel. Centrifugation resulted in 7 mL of plasma. A total of 3 ml of PPP was then removed to achieve a higher platelet concentration. The removal of PPP is optional and aims to achieve a higher platelet concentration according to the manufacturer’s instructions. The rest of the platelets were then resuspended in the remaining plasma to generate PRP by pumping the liquid a few times against the tube wall. A filter sleeve that is provided by the manufacturer was then inserted into the tube until it reached the gel and the PRP passed through the filter and was ready for application. The final PRP concentration was 4 mL and the whole procedure lasted approximately 25 min. Aliquots of 3 ml of PRP were drawn in three separate insulin syringes, and the rest of the volume of PRP was used for analysis including platelet, WBC, and RBC counts. All syringes containing PRP were used within one hour after their preparation.

### 2.4. Skin Wound Creation and PRP Application

After the dorsolateral area of the trunk was clipped and aseptically prepared for surgery, the animals were placed in ventral recumbency, and the wound defects were marked using a sterile skin marker and a millimeter ruler. Three matching 2 × 2 cm square, full- thickness wounds were drawn on each side of the dorsal midline. The extent of wounds was based on previous experimental studies performed in cats [3,49]. The wounds were 3 cm apart from each other and 3 cm away from the dorsal midline (Figure 1). The most cranial wound was marked caudal to the caudal border of the scapula, and the most caudal was far enough from the iliac wing.

The wounds were created using a #15 scalpel blade, and they were full thickness including the excision of the panniculus muscle and subcutaneous tissue until the thoracolumbar fascia was evident. Four simple, interrupted sutures with 3-0 polyamide were placed in the wound corners between the skin and the underlying fascia to avoid slipping or movement from the adjacent tissues. The three wounds on the same side were randomly used as controls, and the other three were used for the application of PRP.

The three wounds on one side, either left or right, were randomly selected for the treatment with the PRP that had been previously prepared, using a computer program (random number generator) (Table 1). In the four corners of each wound on the treatment side (in three wounds in total), 1 mL of the PRP solution was injected and divided into equal doses for each square.

On day 0, a specimen of normal tissue was obtained for histological examination and MMP measurement. The first cranial wounds were used for laser Doppler flowmetry (LDF) measurements and photoplanimetry on days 0, 7, 14, and 25, and they were also used for biopsies for histologic evaluation and MMP measurement on day 25. The two middle wounds were used for biopsies for histologic evaluation on day 7, and the two caudal wounds were used for biopsies for histologic evaluation and MMP measurement on day 14.

### 2.5. Postoperative Care

Postoperatively, the wounds were covered with a bandage consisting of three layers. For the first layer, a non-adherent pad (Cosmopor absorbent adhesive dressing, Hartmann, Germany) with adhesive edges was placed. The second layer included a cotton role gauge (Rolta soft padding bandage, Hartmann, Germany) which was placed around the trunk, crossing around the forelimbs. In the same way, a cohesive, non-slip bandage (Peha-haft self-adhesive bandage, Hartmann, Germany) that adhered to itself was placed around the trunk. Finally, an Elizabeth collar was placed on all cats.

The cats were placed in the recovery room until they were awake and their body temperature was at normal levels. All cats were then transported to their individual cages where they were administered buprenorphine (0.02 mg/kg) intramuscularly 6 h after premedication. Analgesia was continued with the administration of tramadol (1 mg/kg) (Tramal; Vianex, Athens, Greece), subcutaneously, twice a day for 5 days. If a cat was considered to be in pain for more than 5 days according to the Glasgow pain scale, based on the everyday evaluation and clinical examination, tramadol was continued until signs of pain disappeared. Rescue analgesia was achieved by the administration of pethidine (pethidine hydrochloride; Monico SPA, Venice, Italy) [3 mg/kg] intramuscularly.

The bandage was changed daily for the next 25 days, and the wounds were evaluated by the same clinician. All cats were sedated for the bandage change using dexmedetomidine 0.04 mg/kg ΙΜ (DexDomitor; Orion Corporation, Espoo, Finland). Atipamezole (Alzane; Zoetis, Hellas S.A, Acharnes, Greece) was used to reverse the sedation. The duration of this process was approximately 30 min.

### 2.6. Evaluation of Wound Healing

The evaluation of wounds was based on both noninvasive and invasive methods. The noninvasive methods included macroscopic clinical evaluation, LDF, and photoplanimetry, while the invasive included biopsies that were obtained for histologic evaluation and MMP measurement.

### 2.7. Macroscopic Clinical Evaluation

Clinical evaluation was performed daily for 25 days while the cats were sedated and was performed blindly by the same clinician. It included visual inspection of the wounds and determination of the first-day granulation tissue appeared in the cranial wounds, as well as the day when the wound was filled with granulation tissue.

### 2.8. Laser Doppler Flowmetry

For measuring tissue perfusion, LDF was performed on days 0, 7, 14, and 25 from the cranial wounds. The probe of the laser Doppler velocimeter was placed perpendicular and in contact with the wound site according to the manufacturer’s recommendations (Laserflo BPM2, Vasomedics, St. Paul, MN, USA), and three continuous measurements with 5 s intervals were obtained and recorded from the cranial wounds. The mean value of the three recordings was calculated.

### 2.9. Photoplanimetry

Photoplanimetry was performed to evaluate the surface of the open wounds in all cats on days 0, 7, 14, and 25. All wounds were photographed with a sterile ruler on their edges using a digital camera with an mm ruler in the image frame. The surface between the normal skin and the wound was the total wound area, and the area between these two margins was recorded as the area of epithelization. The area within the margin of the advancing epithelium was considered the area of the unhealed wound [9] (Figure 1). Measurements of these surfaces were performed blindly by the same clinician using planimetry software (Rasband, W.S., ImageJ, U.S. National Institutes of Health, Bethesda, MD, USA, https://imagej.nih.gov/ij/ (accessed on 19 July 2022), 1997-2018. The % epithelization, % contraction, and % total wound healing were measured on days 7, 14, and 25 according to the formulas used by Bohling et al., 2006 [3], as follows:% epithelialization day *n* = area of epithelium day *n*/total wound area day *n* × 100(1)
% wound contraction day *n* = 100 − % total wound day *n* (% total wound day *n* = total wound area day
*n*/original wound area day 0 × 100)(2)
% total wound healing day *n* = 100 − % open wound day *n* (% open wound day *n* = open wound area day*n*/original wound area day 0(3)

### 2.10. Histologic Evaluation

Tissues for histologic evaluation were obtained on days 0, 7, 14, and 25 using a 4 mm skin biopsy punch, from the wound edges. On day 0, a biopsy was obtained from normal skin to be used as a control specimen. All specimens were formalin-fixed and paraffin-embedded and sectioned at 5 μm, followed by hematoxylin –eosin staining. All samples were labeled and blindly evaluated by two investigators using an electron microscope. The parameters that were evaluated included inflammatory cell infiltration, edema, collagen production, angiogenesis, and epidermal thickness.

Inflammatory cell infiltration was evaluated by measuring the total number of inflammatory cells (neutrophils, lymphocytes, plasma cells, macrophages, eosinophils, and mast cells) in five high-power fields (HPF) per section. The scoring system included four degrees, with <3 cells HPF considered as the absence of inflammation, and 3–10, 11–30, and >30 cells HPF considered as mild, moderate, and severe inflammation, respectively, with a scoring system from 1 to 4, respectively.

Edema evaluation was based on the existence of material without cells that were not stained or mildly stained between epithelial cells and collagen fibers. The scoring system included four degrees as follows: 1 = the absence of material, 2 = the material mildly separated epithelial cells and collagen fibers, 3 = separation of 30–50 μm, and 4 = separation > 50 μm.

The collagen production scoring system was based on the density of collagen fibers as follows: 1 = the absence of collagen fibers, 2 = a small number of collagen fibers that separated the fibroblasts, 3 = a great density of collagen fibers between the fibroblasts, and 4 = the extensive separation of fibroblasts from the collagen fibers.

The angiogenesis scoring system was evaluated as follows: 1 = an absence or <3 new capillaries HPF, 2 = 3–10 new capillaries, 3 = 11–30 new capillaries, and 4 = >30 new capillaries.

The epidermal thickness scoring system was based on measurement of the thickness on day 25 in five different fields and comparison with the epidermal thickness measured from the control skin on day 0. The scoring system was evaluated as follows: 1 = thickness similar to that of the control, 2 = thickness slightly increased, 3 = thickness moderately increased, and 4 = thickness markedly increased.

### 2.11. Metalloproteinases-2 and -9, and TIMP-1 mRNA Expression

Specimens for MMP evaluation were obtained on days 0, 14, and 25. The specimens that had been harvested were frozen in liquid nitrogen and stored at −80 °C. Real-time qPCR was performed to determine the expression of MMP-2, MMP-9, and TIMP-1. The tryptophan 50-monooxygenase activation protein zeta isoform (YWHAZ) reference gene was selected as an appropriate housekeeping gene according to previous studies [51,52,53]. RNA extraction from biopsy specimens was performed according to Konstantinidis et al., 2021 [54], and reverse-transcription was performed with the SuperScript III first-strand synthesis system (Invitrogen), according to the manufacturer’s instructions. Specific primers were designed, based on the feline GenBank sequences and for each respective target: TIMP1 (sense: 5′-TGGCTGCGAAGAATGCACCGTAT-3′ and antisense: 5′-CTGGAAACCCTTGTCAGTGCCTGT-3′); MMP9 (sense: 5′-CGCACGACATCTTTCAGTTCCA-3′ and antisense: 5′-CCGAGAACTCACACGCCAATA-3′); MMP2 (sense: 5′-GGGTGACCTTGACCAGAGCACGAT-3′ and antisense: 5′-GGTCCAGATCAGGCGTGTAGCCAAT-3′); YWHAZ (sense: 5′-ACAAAGACAGCACGCTAATAATGC-3′ and antisense: 5′-CTTCAGCTTCATCTCCTTGGGTAT-3′) [52].

Primer pair specificities were confirmed by sequencing each respective amplicon. The qPCR reaction (20 μL) was comprised of 1 × PCR buffer (Invitrogen, Carlsbad, CA, USA), 0.2 μM of each primer, 0.2 mM of each dNTP, 3 mM MgCl_2_, 3 U of Platinum^TM^ Taq DNA Polymerase (Invitrogen, Carlsbad, CA, USA), 1 × EvaGreen^TM^ dye (Biotium, Hayward, CA, USA), and 2 μL of cDNA. The following thermal cycling conditions were applied: 94 °C for 3 min, followed by 47 cycles in 2 steps: (a) denaturation at 94 °C/20 s and (b) annealing-extension at 60 °C/45 s for the MMP-2 and TIMP-1 primers or at 57 °C for the MMP-9 and YWHAZ primers, respectively. Fluorescence was recorded at the end of each cycle. After the completion of cycles, a melting curve was generated by heating from 75 °C to 90 °C in increments of 0.2 °C/6 s. Reactions were carried out on a CFX96 Touch^TM^ Real-Time PCR Detection System (Bio-Rad Laboratories, Hercules, CA, USA). Reaction efficiencies were determined for each primer set, using 10-fold serial dilutions (2 × 10^7^–2 × 10^1^ copies/reaction) of plasmids ligated with each amplicon, and were found to be close to 100%. A melting curve analysis did not show nonspecific amplicons in any of the reactions. All RNA extracts were examined in triplicate. Target gene expression levels were determined by the comparative threshold cycle (ΔCT) method [55]. Data were expressed as fold changes over control samples given by 2−ΔΔCT.

### 2.12. Statistical Analysis

Data are presented as mean ± SD. For the variables LDF, epithelialization percentage, contraction percentage, total wound healing percentage, and MMP-2, MMP-9, and TIMP-1 activity, a double repeated measures analysis of variance were performed, with these variables as a dependent, the cats (eight cats) as the sampling unit, and the wounds (two wounds) and the sampling days (0, 7, 14, 25 days or 7, 14, 25 days) as independent variables. For the variables of cell infiltration, edema score, collagen production, epidermal thickness, and angiogenesis, the non-parametric Wilcoxon statistical test was performed.

The statistical analysis was performed using IBM SPSS Statistics v26.0 (IBM, Armonk, NY, USA). The observed significance levels (*p*-values) of statistical tests were predetermined at α = 0.05 (*p* ≤ 0.05) and were estimated by the Monte-Carlo simulation method. On the basis of feline wound healing in previous studies, we determined that a group size of eight would provide at least 80% power to detect differences at (*p* < 0.05) [1,3,56].

## 3. Results

### 3.1. PRP Analysis

The platelet counts in PRP were increased in comparison with the blood platelet concentration. A 2–8.2-fold increase in PRP concentration was calculated. The mean PRP platelet concentration was 1,181,750 ± 702,938 K/μL, while the mean blood platelet concentration was 279,750 ± 55,140 K/μL (Table 2). The mean RBC volume after centrifugation was 0.0 M/μL, and the WBC volume after centrifugation was 0.73 ± 0.4 K/μL.

### 3.2. Clinical Evaluation

No significant difference was recorded in the first appearance of granulation tissue between the control (5.625 ± 0.48 days) and PRP (5.0 ± 0.86 days) groups (*p* = 0.117). The mean time needed for full coverage with granulation tissue in the PRP group was significantly less (10.88 ± 1.26 days) than the mean time in the control group (12.13 ± 1.28 days) (*p* = 0.08).

### 3.3. Photoplanimetry

#### 3.3.1. Epithelialization

A significant difference was found in the epithelialization percentage in both the control and treatment groups during the study, with the percentage increasing from day 0 to day 25 (*p* < 0.001). However, no significant difference was observed between the control and treatment groups on days 7, 14, and 25 (*p* = 0.823). No significant treatment–control interactions were reported (*p* = 0.973).

#### 3.3.2. Contraction

A significant difference was recorded in both groups on all measurement days, with the contraction percentage rising from day 0 to day 25 (*p* < 0.001). The mean contraction percentage of the PRP group (52.2%) was significantly higher than that of the control group (40.75%) during all examination days (*p* = 0.002). On day 14, a rise in contraction was observed in both the control and PRP groups, with the percentage almost tripling for the PRP group (59.251 ± 14.834) and increasing more than eight times for the control group (42.73 ± 16.028). On day 25, the contraction percentage was up to 70% in both groups. No significant treatment–control interactions were reported (*p* = 0.119).

#### 3.3.3. Total Wound Healing

Th mean percentage of total wound healing in the PRP group (68.51%) was greater than that of the control group (61.19%) (*p* = 0.006). On day 7, the mean total healing in the treatment group was greater (30.887 ± 9.614%) than in the control group (16.462 ± 11.612%) [*p* = 0.006]. The total wound healing percentage increased in both groups at all measurement times from day 0 to day 25 (*p* < 0.001). Total wound healing reached over 95% in both groups on day 25. An interaction was also noted between the two groups (*p* < 0.05) indicating that PRP application enhances wound healing.

Epithelialization, contraction, and total wound healing measurements are presented in Table 3. All figures of wounds’ healing during all measurement times are included in Figure 2 and Figure 3.

### 3.4. Laser Doppler flowmetry

Blood flow was higher in the PRP group on days 7, 14, and 25. In the PRP group, an increase in flowmetry was recorded from day 0 to day 7, followed by a decrease until day 25, whereas in the control group, the increase continued until day 14. A peak value in the PRP group was recorded on day 7, while in the control group, the peak was recorded on day 14. A significant difference was found between days 0 and 14 (*p* = 0.003) and between 0 and 25 (*p* = 0.006) in both groups. No interaction was noted between the days of examination or the different treatment groups (*p* = 0.207). A significant difference was noted in Doppler flowmetry during all examination days between the groups, with the PRP group having a higher measurement (*p* < 0.05). All LDF measurements are presented in Table 4.

### 3.5. Histologic Evaluation

No statistical difference was recorded in cell infiltration between the control and PRP groups on day 7 (*p* = 0.753), 14 (*p* = 0.500), or 25 (*p* = 0.440). A significant difference was found only in the control group at the different measurement times. More specifically, the inflammation was different between day 0 and day 7, when the inflammation was evaluated as absent and mild, respectively (*p* = 0.017); between day 0 and day 14, when the inflammation was evaluated as moderate to severe (*p* = 0.007); and between day 0 and day 25, when the inflammation was mild to moderate (*p* = 0.01).

The platelet-rich plasma and control groups did not differ significantly in edema scoring on any of the measurement days (*p* = 0.501 on day 7, *p* = 1 on day 14). Edema was increased in both groups from day 0 to day 7, followed by a decrease on day 14, and reaching the absence of edema on day 25. In the control group, the edema scoring differed significantly from day 0 to day 7 (*p* = 0.016), and from day 7 to day 25 (*p* = 0.016). In the PRP group edema, the scoring differed significantly from day 7 to day 25 (*p* = 0.018).

No difference existed in collagen production between the two groups on any of the measurement days (*p* = 1 on day 7, *p* = 0.496 on day 14, *p* = 1 on day 25). Collagen production showed a decrease in both groups from day 0 to day 7, followed by an increase on day 14 in the control and PRP groups, which continued to rise in both groups on day 25. In the PRP group, collagen production was significantly higher from day 7 to day 14 (*p* = 0.011), and from day 7 to day 25 (*p* = 0.017). Collagen production in the control group differed significantly from day 0 to day 25 (*p* = 0.017), and from day 7 to day 25 (*p* = 0.017).

Angiogenesis showed an increase from day 0 to day 7 in both groups, with the mean score being almost double in the PRP group (2.38 ± 0.324) compared to the control group (1.38 ± 0.263). Angiogenesis continued rising from day 7 to day 14, followed by a decline in both groups on day 25. However, no significant difference was noted in the number of new vessels between the two groups on any of the measurement days (*p* = 0.065 on day 7, *p* = 0.249 on day 14, *p* = 0.126 on day 25). A significant difference was recorded in the control group between day 0 and day 14 (*p* = 0.015), between day 0 and day 25 (*p* = 0.009), and between day 7 and day 14 (*p* = 0.0031). No difference was recorded in the thickness of the new epidermis on day 25 (*p* = 0.628) between the groups. All histologic measurements are presented in Table 5.

### 3.6. Metalloproteinases

#### 3.6.1. MMP-2

A significant difference was noted in MMP-2 expression in both groups during all measurement times (*p* = 0.028). The metalloproteinase-2 activity on day 14 was higher than on day 0 in both groups (*p* = 0.035) and was decreased from day 14 to day 25 (*p* = 0.022). On day 14, the MMP-2 activity in the PRP group was increased 8.313 (±3.871) times compared to day 0, and on day 25, the MMP-2 activity was 0.86 (±0.29) times higher than on day 0. In the control group, the MMP-2 activity on day 14 was 6.74 (±3.08) times higher than on day 0, and on day 25, it was 0.74 (±0.25) times higher than on day 0. No significant difference was observed in measurements of MMP-2 activity between the two groups (*p* = 0.734), and no interaction was noted between the groups (*p* = 0.768).

#### 3.6.2. MMP-9

No significant difference was recorded in MMP-9 activity measurements during the measurement times (*p* = 0.088). Similarly, no difference was noted between the two groups on any of the examination days (*p* = 0.511) and no interaction between the two groups was evident (*p* = 0.346). The metalloproteinase-9 activity on day 14 in the PRP group was 37.86 (±25.02) times higher than on day 0, and on day 25, it was 9.17 (±3.98) times higher than on day 0. In the control group, MMP-9 was 13.56 (±6.8) times higher than on day 0, and on day 25, it was 14.53 (±8.57) times higher compared to day 0.

#### 3.6.3. TIMP-1

A significant difference was observed in TIMP-1 activity during all the examination days in both groups (*p* = 0.042). The tissue inhibitor metalloproteinase-1 activity increased 15.13 (±7.09) times in the PRP group from day 0 to day 14, and 1.4 (±0.27) times from day 0 to day 25. In the control group, the TIMP-1 activity on day 14 was 17.2 (±7.89) times higher compared to day 0, and on day 25, the TIMP-1 activity was 1.16 (±0.29) higher compared to day 0. No significant difference was found between the two groups (*p* = 0.841), and no interaction was observed between the two groups (*p* = 0.81). All MMP and TIMP-1 measurements are presented in Table 6.

## 4. Discussion

In the study presented here, we reported for the first time in laboratory cats that the mean time for full coverage with granulation tissue in the PRP group was significantly less than that in the control group. The mean contraction percentage and the mean total wound healing percentage were significantly greater in the PRP than in the control group. Finally, the mean LDF in the PRP group was higher than that of the control group during all examination days.

Platelet-rich plasma concentration has been widely used in veterinary medicine as a cost-effective and simple method, with many reports performed in dogs, rabbits, and horses [15,18,46,48,57,58,59]. In our study, PRP was locally injected into surgically created open wounds in laboratory cats and to the authors’ knowledge, this is the first study with PRP application in cats. Our initial hypothesis that PRP enhances wound healing in open wounds in laboratory cats is supported by the results of the present study.

Platelet-rich plasma concentration is considered to be any blood product with a platelet concentration over the baseline, with the commercially available systems usually achieving a 2–5-fold increase [21,22,23]. A product is considered as PRP if it has a platelet concentration 2–5 times the platelet concentration in whole blood [20,21]. In humans, there are studies where it is considered that platelet concentration in PRP should be 4–7 times the peripheral blood concentration to have a beneficial effect [60]. However, there are other studies that have proven the beneficial effect of PRP with the platelet concentration demonstrating a 1–6-fold increase compared to whole blood [61,62,63,64], or studies where an excessive increase in platelets in PRP seemed harmful [65,66]. Weibrich et al. also suggest that different concentrations are needed for different tissues [67]. Currently, an optimal PRP concentration is largely unsubstantiated and the beneficial effect of PRP associated with platelet concentration should be further investigated [20,68,69]. In veterinary medicine, there is no optimal platelet concentration supported by the literature [70]. In the present study, four of eight cats exhibited a platelet increase in PRP of at least 2-fold compared to whole blood. The product had the characteristics of PRP including the platelet increase over the baseline, RBCs, and WBCs reduction according to the manufacturer’s standards so the product was considered suitable for the application. It is also known that platelets are not the only important parameter associated with the beneficial effect of PRP [70]. In our study, the platelet concentration was 2–8.2-fold higher compared with the baseline (Table 2) with a mean increase of 4.1 times. This finding is in contrast with the two reports that evaluated the characteristics of feline PRP [58,59]. Chun et al. (2020) reported a 2.5-fold increase compared to the baseline using a commercially available PRP system [70]. Ferrari and Schwartz (2020) used two different commercially PRP systems and only in one system an increase of 187% was recorded while in the second system the concentration of platelets in PRP was decreased [21]. The underlying cause for low platelets in PRP might be due to an error of the analyzer to identify the platelets or due to the different PRP preparation methods or it might be species specific or to the phenomenon of pseudothrombocytopenia that is not rare in cats [21,70]. In contrast in dogs, a rise of up to 21-fold in platelets in PRP has been recorded [48]. It is essential for the PRP characteristics to be evaluated before the topical application because there are differences between the different commercial systems [21,25]. The features that are usually evaluated before PRP application are the number of platelets, WBCs, and RBCs [20,45,48,59,71]. In our study a commercially available human kit was elected for the preparation of PRP that has never been validated in cats. There are only two studies that have performed validation of PRP in cats using different systems. In these studies, the mean whole blood and PRP product platelet, RBCs, and WBCs concentration were determined in the different systems [21,70]. None of these studies evaluated growth factors concentration. The growth factors that play a fundamental role in wound healing are not usually evaluated before PRP application [20]. In dogs, there is only a report evaluating the growth factors in canine autologous conditioned plasma [72]. The concentration of growth factors is also not always related to the rise of platelets in PRP [36,67,73]. No studies performing validation of the Tropocells system in cats appeared in the literature. However, in the present study we have analyzed the main characteristics of PRP and according to our findings, the PRP features in the cats of the present study were similar to the data given by the manufacturer.

Platelet-rich plasma also contains RBCs, WBCs, and growth factors that are released from platelet α-granules [23,74]. Red blood cells and WBCs concentration in PRP seem to play an important role to exert its beneficial effect [24]. Ehrenfest (2009) described a classification of PRP based on its concentration in different cells especially WBCs including pure platelet-rich plasma (P-PRP), the leukocyte- and platelet-rich plasma (L-PRP), the pure platelet-rich fibrin (P-PRF), and the leukocyte- and platelet-rich fibrin (L-PRF) [75]. Generally, it is considered that the number of WBCs and RBCs should be reduced because they stimulate inflammatory response [24,76,77]. In contrast, studies that support the use of L-PRP due to the presence of leucocytes in the inflammatory phase of wound healing were also published [76,78,79,80]. In veterinary medicine, L-PRP has been used in 17 dogs with aural hematomas that were resolved completely after the injection of L-PRP [81]. In our study, a commercially available system was used (Tropocells) to prepare PRP which had a very low number of WBCs, no RBCs, and a high number of monocytes. This type of PRP may be classified as a leucocyte poor PRP according to the classification by Le et al., 2018, having a concentration of leucocytes below the baseline [82]. In general, PRP products differ in the method for PRP preparation, the volume of blood needed, the amount of PRP that is prepared, the rise in the platelet concentration compared to the baseline, and finally the concentration of leucocytes and RBCs [25,83]. The system used in the present study was a single centrifugation system. There are many commercially available systems using a single centrifugation including Arthrex ACP Double Syringe (Arthrex, Naples, FL, USA), Arthrex Angel System (Arthrex, Naples, FL, USA), RegenKit A-PRP (RegenLab, Lausanne, Switzerland), PRGF/Endoret (BTI, Alava, Spain), Magellan (Arteriocyte Medical Systems, Hopkinton, MA, USA), etc. [84]. The PRP volume obtained is also different with the different devices.

In our study, there was no significant difference in the first appearance of granulation tissue between the control and treatment groups. The granulation tissue first appeared in the wound edges on day 5 in the PRP group and on day 5.6 in the control group. Our findings can be favorably compared with those of Bohling et al., 2004 [85], where granulation tissue was obvious in cats with full-thickness wounds within 6 days. In our study, a significant difference was observed in full coverage of the bottom of the wound with granulation tissue between the PRP and control groups. We believe that this difference occurred because the activation of platelets in PRP releases growth factors including PDGF, TGF-β, VEGF, FGF, EGF, and IGF which have been proven to be increased 3–7 times in PRP compared to peripheral blood [17]. All these growth factors play an important role during wound healing [86]. Platelet-derived growth factor has been used in vivo in mice increasing angiogenesis and the number of fibroblasts, and enhancing wound healing [87]. It has also been reported to play a role in the maturation of granulation tissue, collagen, and fibroblast production, improving wound healing in dogs [88]. We found that the application of PRP contributed to the earlier coverage of wounds with granulation tissue, in contrast with the results of Bohling et al. in 2004; in open wounds in cats after the removal of subcutaneous tissue, the coverage of wounds with granulation tissue was completed in 18 days (*p* = 0.042) with only five cats having a complete coverage on day 21 [85].

In the present study, the percentage of epithelialization, contraction, and total wound healing was evaluated using Image J software that has previously been used in other studies [54,89]. The percentage of epithelialization showed no significant difference between the control and treatment groups on all days of examination in the present study. However, the percentage of epithelialization in the present study is in contrast with the study of Bohling et al. (2006) in which only one cat had epithelialization > 50%; in our study, the mean percentage of epithelialization reached over 80% in both the control and treatment groups [3]. These differences might be attributed to individual differences between the animals, the PRP application, the earlier coverage of the wound with granulation tissue in our study, or to the longer duration of our study which lasted 25 days while Bohling’s study lasted 21 days. Our findings agree with the study of Tsioli et al. (2016) in which they investigated the results of two occlusive, hydrocolloid dressings in second-intention wound healing in cats and reported an epithelialization percentage of over 85% on day 21 in all treatment groups [56]. In dogs, PRP has been proven to have a beneficial role in epithelialization percentage [15].

In the study presented here, a significant difference in the percentage of contraction was observed between the control and PRP groups during all measurement times, with the PRP group having a higher percentage of contraction (*p* < 0.002). In cats, contraction is the main mechanism for wound healing, whereas in dogs, wound healing is mainly based on epithelialization [85]. Contraction proceeds slowly in cats within the first 7 days of healing [85]. In our control group, the percentage of contraction was 5.775% on day 7, increasing up to 40% until day 14. In the PRP group, the percentage of contraction was almost four times greater compared to the control group on day 7. This difference could be due to the earlier appearance of granulation tissue. Granulation tissue is a source of fibroblasts which are further differentiated into myofibroblasts secreting type I and III collagen [71,90]. Myofibroblasts are the main cells during contraction due to their ability to extend and retract [91]. However, in the present study, no difference was noted in the first appearance of granulation tissue. We believe that the increased concentration of PDGF from the platelet degranulation in the PRP group might be the reason for the difference in contraction, as PDGF is mostly involved in this process [90]. The rise in contraction in both the control and PRP groups from day 7 to day 14 agrees with the findings of Bohling et al., 2004, where a more rapid contraction was found in cats compared to dogs from day 7 [85].

Platelet-rich plasma was also proven to enhance total wound healing during all examination days in our study. A significant difference was found between the control and PRP groups at all measurement times (*p* = 0.006). On day 7 specifically, the percentage of total wound healing in the PRP group was almost double in comparison with the control group—with percentages of 30.886% and 14.462%, respectively. This finding might be attributed to a larger number of new vessels appearing in the PRP group. On day 7, angiogenesis was found to be almost doubled in the PRP group in comparison with the control group. However, no significant differences were recorded (*p* = 0.065). The better healing percentage in the PRP group might be attributed to the secretions of growth factors for at least 4 days [78]. The wound healing percentage was increased during our study in both the PRP and control groups, reaching almost 97% on day 25. In dogs, there are studies indicating a higher rate of total wound healing in PRP groups compared to controls [48,71]. Tambella et al. (2014) also found a quicker wound healing in chronic decubital ulcers in 18 dogs in which PRP was injected [18]. However, a systematic review and metanalysis in 2018 have shown the beneficial role of PRP for wound treatment in dogs even though the healing time was not enhanced [92]. In our study, despite the subcutaneous tissue removal, the percentages of epithelialization, contraction, and total wound healing were comparable with the percentages of wounds with intact subcutis in dogs in Bohling et al.’s study (2006) [3]. Gemignani et al. (2017) reported an epithelialization percentage of 80% and contraction of 90% on day 10, and a wound size reduction of 50% on day 4 in a cat with an open wound after the application of canine PRP [49]. However, this is a case report with the use of heterologous PRP that needs to be further investigated in a larger sample of cats.

Cutaneous flowmetry in our study showed a higher blood flow in the PRP group at all measurement times compared to the control group (*p* = 0.05). This difference might be due to the VEGF secretion in the PRP group resulting in vessel dilation, because a difference in angiogenesis was not detected on any of the examination days [48]. The LDF is not a real measurement of blood flow in the microcirculation, but an expression of capillary blood flux [93]. Manning et al. (1991) reported the variability of blood flow in different sides and species [93]. Some of the main disadvantages of this non-invasive method are the difficulty of calibration of the machine, the movement artifact that can affect the results, and some variables that can affect the measurements including age, sex, anatomical sides, individual variations, drugs, or estrus cycle [94]. The anesthetic drugs that were used might have affected blood flow. In mice, it was reported that isoflurane and acepromazine increase blood flow, and dexmedetomidine initially results in a decrease followed by a rise in perfusion after 5 min [95]. However, this is a factor that does not seem to affect the difference between the two groups in our study, because PRP and control groups were created in the same cat. The blood flow in the PRP group seems to increase until day 7 followed by a decrease until day 25, whereas in the control group, the increase continues until day 14. This finding may be attributed to the quicker healing of the PRP group in comparison with the control group, which was still in a higher metabolic rate due to slower healing [3].

No significant difference was detected in inflammatory cell infiltration between the PRP and control groups on all measurement days of the study presented here. The inflammation showed an increase from day 0 to day 14 in both groups, followed by a decrease until day 25. The inflammation phase, the first phase of wound healing, usually lasts about 5 days [90,96]. Some of the inflammatory cells including macrophages which secrete PDGF and TGF-β growth factors also play a fundamental role in the proliferative phase [90,97,98], and the absence of these cells may retard wound healing [98]. There are studies in which the inflammation phase can last up to 2 weeks [91]. However, the normal inflammation phase should not be confused with the inflammation that is a result of infection and can be distinguished by gross examination [97]. Platelet-rich plasma has been proven to be beneficial in infected wounds in rats, reducing the bacteria number and increasing wound healing [99]. In another study, PRP was proven to have anti-inflammatory properties in MRSA infection in six dogs, showing better results in epithelialization and granulation tissue formation compared to controls [100]. Edema scoring showed no difference between the PRP and the control group in the present study. The edema score in both groups showed an increase until day 7, followed by a decrease over the subsequent days until no edema was observed on day 25. In both groups of the present study, this decrease in edema scoring was favorably compared to the increased angiogenesis scoring.

Angiogenesis was not significantly higher in the PRP group—although on day 7, a nonsignificant increase was noted. Angiogenesis takes place during the proliferative phase of healing, combined with granulation tissue formation and re-epithelialization [90]. The first appearance of granulation tissue also showed a nonsignificant difference between the groups in the present study. Karayannopoulou et al. (2015) found no difference in angiogenesis in wound healing in dogs after the application of PRP [48]. Our findings are in contrast with other studies in which PRP improved angiogenesis in open wounds or flaps due to the release of high levels of VEGF which is considered the main growth factor in the angiogenesis process [101,102].

No significant difference was found between the two groups in either the collagen production score or in the epidermis thickness on day 25. There are studies that have shown that PRP improves collagen organization which plays an important role in wound strength [48,58,71,101]. The determination of the ratio of collagen type I and III could have aided in the evaluation of the effect of PRP in wound healing.

Metalloproteinases are known to play a fundamental role during the different phases of wound healing. This is the first study evaluating MMPs and TIMPs during the wound-healing process in small animals, with only one study in dogs reporting MMP-9 activity in wound fluid [15]. The method elected in the present study was RT-PCR. In other veterinary studies, gelatin zymography has been used as a semiquantitative method [12,15]. In both MMPs of our study (MMP-2 and MMP-9), an increase in activity was recorded until day 14, followed by a decline until day 25. Tissue inhibitor metalloproteinase-1 regulates the expression of MMPs. The role of MMPs is important because it has been found that the overexpression of MMPs may delay wound healing [7]. In our study, on day 14, TIMP-1 activation was increased to limit the production of MMPs, followed by a decrease on day 25 when both MMP-2 and MMP-9 activation were almost at the baseline. It has been reported that in chronic wounds, the expression of TIMPs is increased [7].

In the present study, no significant difference was found in MMP-2 or -9, or TIMP-1 between the control and PRP groups. MMP-2 expression has been found to be correlated with keratinocytes’ increase in wound edges, accelerating cell migration [6]. However, the role of MMP-2 remains unclear. Metalloproteinase-9 has a role in keratinocyte migration and reepithelialization [9,103,104]. Both metalloproteinases also have a role in angiogenesis [8,104]. In the present study, MMP-9 was almost tripled on day 14 in the PRP group compared to the control. This finding might explain the peak of angiogenesis on day 14 in the PRP group due to the activation of angiogenic cytokines such as TNF-a and VEGF [8,104]. It has also been found that mice with a lack of MMP-9 show a delay in wound healing [105]. In the present study, all wounds in both the control and PRP groups were healed at a percentage over 96% on day 25, with no wound showing an impaired wound healing. The overexpression of MMPs can lead to impaired wound healing and an increase in their activity is noted in nonhealing wounds or in diabetic patients [24,106,107]. In our study, no difference was found in MMP or TIMP-1 concentrations between the PRP and control groups, in contrast with Farghali et al. where a significant difference was recorded in MMP-9 activity during the second and third week of healing [15]. Our study reported for the first time in the literature the activity of MMPs and their inhibitor during all phases of wound healing in cats.

In the present study, three females and five male cats were included. The main factors that can affect the process of cutaneous wound healing are local and systemic factors. Systemic factors include primary immunodeficiencies, the presence of cancer, or age—as healing in older animals is different compared to young animals [108]. Sex is a factor that could have affected the LDF [94]. However, in our experiment the control and treatment groups were in the same animal, so the factor of sex did not affect our results. Similar experimental studies have not recorded sex as a factor affecting the results of the healing process [3,48,56].

In our study, we elected to apply PRP intralesionally, a technique first described by Dionysiou et al. in 2012 [44]. The intralesional technique was elected because a single application was preferred compared to PRP gel which would necessitate daily applications. Local PRP gel application may also activate an inflammatory process causing a production of fluid that may delay wound contraction, and the risk of slipping away from the wound during dressing changes is reduced [44,58,71].

The limitations of the present study include the use of PRP in surgically created wounds in normal laboratory cats. The results might be different if the study was performed in a clinical setting, in infected wounds, or if using a different PRP system. Growth factor evaluation may have helped in a better interpretation of our results. This will justify the cautious interpretation of our results.

## 5. Conclusions

To our knowledge, this is the first experimental study to evaluate the effect of PRP applied once daily in wound healing in cats, and the results are encouraging. It seems that PRP intralesional administration improves contraction, total wound healing, and tissue perfusion in cutaneous deficits with the subcutaneous tissue removed in cats. Further investigations are needed to determine the concentration of growth factors in feline PRP and their role in different stages of feline wound healing.

## Data Availability

Not applicable.

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
