# Peer review of "Locally Injected Autologous Platelet-Rich Plasma Improves Cutaneous Wound Healing in Cats"

_animals, 2022, doi:10.3390/ani12151993_

Round 1
Reviewer 1 Report
The authors consolidated important revisions in the manuscript.
I suggest publication in the manuscript in the present format.
Author Response
The authors would like to thank the reviwer for the comments that have improved the quality of the manuscript. The english language was checked.
Reviewer 2 Report
Your abstract seems to say that PRP lengthened the time for granulation coverage. I think it shortened the time.
I would also say something about why you felt this study would not have been effective as a larger observational study. The interventions are a bit harsh from the humane standpoint. You might also say how you decided on this design, and minimized animal use.
Author Response
The authors corrected the line that says that PRP lenghthened the time for granulation tissue coverage (New line 44). A comment has been also added to expain the design and the election of the number of the laboratory cats (New line 132-133.
Thanks for your observations
This manuscript is a resubmission of an earlier submission. The following is a list of the peer review reports and author responses from that submission.
Round 1
Reviewer 1 Report
The aim of this study was to investigate the efficacy of PRP infiltration in open wounds in cats. It was found that wounds infiltrated with PRP showed a more rapid total wound healing and better perfusion.
The article is well written and the introduction is complete, but unfortunately I have great concerns about how the PRP was produced and how the experimentation was set up.
In fact, in my opinion, there are two key points that unfortunately make the conclusions of this study unreliable: the first concerns the production of PRP, for which a human kit was used that has never been validated in the feline species. And this is a key point because too often in the past PRP which did not have adequate characteristics has been clinically used and this has led to very conflicting results in publications and literature with little scientific value, with a loss of confidence in many potentially valid therapies. Before using PRP in clinical cases it is essential to validate the production method in the species. The second, less severe, is the non-probable extension of the results obtained on experimentally induced surgical wounds in healthy laboratory cats to the lesions that are often found in the feline clinical practice and on which PRP should be directly tested, i.e. traumatic wounds with laceration of the tissues, wound obteined after oncology surgery , fight / bite wounds, autoinduced lesions in allergic cats etc. While the second perplexity can be resolved by modifying the introduction, discussions and conclusions focusing attention only on the regeneration result on skin in laboratory cats without extending it to all types of cats and wounds, the first is not resolvable, since the system used for production was not previously validated in cats, at least by consulting current literature.
Here are my more detailed observations:
line 31: add "experimentaly created" before wound and "laboratory" before cats
line 39: add "laboratory" before cats
line 142: how many gauge did the needle have used? and what syringe was used? These are important data that must be mentioned
lines 147-150: are a repetition of the previous lines (141-145)
line 149: which authomatic analyzer?
lines 150-151: cut and past at line 141, before the sentence "Before the induction..."
line 152: this passage is not clear to me: in the previous paragraph the authors state that 12 ml were taken from the cat, one of which is put into edta for counting, then here instead they say that the ml for the PRP were taken directly with the vacuum tube of the kit. Were two separate withdrawals then made?
line 152: Is this a kit for human use? Has it ever been validated before in the feline species? Are we sure that a single centrifugation is enough to produce a good PRP? the literature on this is not in agreement. Previous studies have shown that there are variations in platelet, leukocyte, and red blood cell concentrations between systems for a given species. Additionally, it has been shown that systems validated for humans may not yield the same product parameters for animals, underlining the fact that the PRP product of one species may not necessarily be representative of the PRP product from another species using the same system (as demonstrated in cat by Ferrari et al, 2020). Before using a kit for the production of PRP for clinical purposes it is imperative, in my opinion, to carry out validation studies of that kit for the species, with a high number of subjects, as good laboratory practices indicate. The great variability of the PRP produced with the kit used in this study in cats is demonstrated by the very different platelet values obtained in the 8 subjects included in the study. Who can guarantee that the platelets did not rupture in the process and that the growth factors were released early in the supernatant? Was a smear performed to assess platelet morphology in PRP? Has the concentration of growth factors been evaluated in the obtained PRP or other platelet integrity parameters, such as P-selectin for example? Perhaps it has been done in previous studies with this kit in cats? Is the PPP been evaluated?
line 153: what anticoagulant is in the kit?
line 157: How much PPP has been removed? how was the quantity to be removed decided? The amount of PPP to be removed is a key step to obtain a good platelet concentration
line 160: what kind of filter is it? was it a step of the method decided by manufacturer?
line 161: I think the authors mean volume, not concentration. In addition, I am amazed by the quantity of 4 ml PRP for other in all subjects, obtained starting from only 11 ml of whole blood. It is a lot compared to previous studies in dogs and cats (Perego et al 2021, Iacopetti et al 2020, Chun et al 2020, Ferrari et al 2020, etc.)
line 180: how was the randomization done? with a computer program? from an operator not involved in the study?
Table 2: in line 90 the authors use the definition of PRP which cites the achievement of a million platelets or a concentration 3 or 4 times higher than the baseline value. Looking at the table in 4 out of 8 cats these conditions are not met. In the discussions you report other parameters for evaluating the PRP. The literature on this topic is still controversial, but surely it is necessary to decide which parameters to consider and if in the introduction one mentions the million platelets or the 3-4 times compared to the baseline, I as a reader expect that the PRP obtained respects these characteristics.
line 478-497: The same authors report an important discussion on evaluating the PRP obtained before use, but in fact, in their study this preliminary evaluation of the kit was not carried out following the good laboratory practices that provide for the validation of a method and the studies they cite show that in cats it is essential to use kits that are validated for the species.
line 510: Are where the data on WBC and RBC in PRP?
Reviewer 2 Report
The aim of the study entitled “Locally injected autologous platelet rich plasma improves cutaneous wound healing in cats” was to record the efficacy of PRP infiltration in open wounds in cats.
The manuscript is well constructed, however an English revision is required.
There is an institutional authorization letter that is in Greek, please include it in English so we can understand. The other letter is neither signed nor stamped. Please provide these documents with the necessary modifications.
Abstract
Please include the meaning of the abbreviation PRP
Materials and Method
How did the authors ensure that the treatment performed on one wound did not influence the contralateral treatment?
Please, what is the reference for the number and extent of the wounds?
The wounds were quite extensive and occupied a good part of the ventral portion of the animal. How did the authors monitor the animals' pain and well-being? Was any pain reliever instituted after the surgery?
- L. 155 - 10 minutes at 1500 rpm - 10 min at 1,500 rpm.
- L. 162 - 25 min.
Results and discussion
- Please include photographs of the evolution of the morphological changes of the wounds.
- The authors used 3 females and 5 males. In relation to the parameters analyzed, was there a difference in relation to the sex of the animals? Please include some comment about it and discuss.